# Lost in Transition: Challenges in the Journey from Pediatric to Adult Care for a Romanian *DMD* Patient

**DOI:** 10.3390/healthcare13070830

**Published:** 2025-04-05

**Authors:** Maria Lupu, Maria-Alexandra Marcu, Diana Anamaria Epure, Oana Aurelia Vladacenco, Emilia Maria Severin, Raluca Ioana Teleanu

**Affiliations:** 1Faculty of Medicine, Carol Davila University of Medicine and Pharmacy, 020021 Bucharest, Romania; maria.lupu@rez.umfcd.ro (M.L.); oana-aurelia.vladacenco@drd.umfcd.ro (O.A.V.); emilia.severin@umfcd.ro (E.M.S.); raluca.teleanu@umfcd.ro (R.I.T.); 2Department of Paediatric Neurology, Dr Victor Gomoiu Children’s Hospital, 022102 Bucharest, Romania; epurediana@gmail.com

**Keywords:** duchenne muscular dystrophy, dystrophin gene, nonsense pathogenic variant, transition

## Abstract

Background: The transition from pediatric to adult care in Duchenne Muscular Dystrophy (*DMD*) is challenging due to the disease’s complexity and the need for lifelong, comprehensive management. In Romania, ongoing efforts aim to enhance multidisciplinary collaboration, though systemic barriers, such as fragmented healthcare services, persist. Nonsense mutations, including those in exon 30 described here, are often associated with more severe disease progression. Methods: We present the case of a 17-year-old Romanian *DMD* patient with a nonsense mutation in exon 30 of the dystrophin gene. The patient received multidisciplinary pediatric care addressing his medical needs, including neuromuscular, respiratory, cardiac, and orthopedic management. Transition readiness was assessed using the Transition Readiness Assessment Questionnaire (TRAQ), and the patient’s perspective on the process was documented. Results: Care followed international standards, but the disease progressed predictably, with gradual loss of ambulation, respiratory decline, and cardiac complications. The TRAQ revealed strengths in communication with healthcare providers but moderate confidence in self-management tasks. From the patient’s perspective, fragmented adult services and difficulty accessing specialized neuromuscular support remain major obstacles, underscoring the importance of early, structured transition planning and patient-centered approaches. Conclusions: Transitioning to adult services requires strong communication between pediatric and adult teams and integration of specialized care. Tailored follow-up plans ensure continuity of care and effective disease management. This case reflects broader needs in similar healthcare contexts, highlighting the necessity of robust transition frameworks to respond to patient-specific challenges and ultimately support long-term quality of life.

## 1. Introduction

Dystrophinopathies are a group of neuromuscular disorders caused by variants in the dystrophin-encoding gene, leading to absent or deficient dystrophin production and progressive muscle degeneration [1,2]. Duchenne Muscular Dystrophy (*DMD*) (OMIM #310200; ORPHA:98896), an X-linked genetic disorder, is the most common and severe form of muscular dystrophy [1,3]. Beyond its primary neuromuscular involvement, *DMD* is a multisystemic disorder that affects the cardiovascular, respiratory, gastrointestinal, endocrine, and central nervous systems [1,4,5]. Given the extensive nature of its clinical manifestations, effective management necessitates a multidisciplinary approach [1,4,5,6].

Since the 1990s, the introduction of steroid therapy, the broader use of ventilation support, comprehensive multidisciplinary care, and the protective administration of angiotensin-converting enzyme (ACE) inhibitors have all contributed to a marked increase in life expectancy, with many patients now living well beyond 40 years of age [1,2,6,7,8].

With ongoing improvements in care and extended life expectancy, the transition from pediatric to adult-focused medicine has become essential for all *DMD* patients. A structured transition plan that anticipates individual complexity and needs remains crucial to maintaining high standards of care [2,7,9,10,11].

Nonsense variants, which comprise around 10–15% of *DMD* cases, introduce a premature stop codon leading to a truncated dystrophin protein [12,13,14]. Their clinical impact varies based on the stop codon’s location [15], but certain positions have been linked to earlier onset of cardiomyopathy, cognitive impairments, and accelerated muscle deterioration [16,17,18]. Such genotypic differences highlight the importance of tailored multidisciplinary management, ensuring each patient’s unique profile is addressed across specialties [5,15,16,17,18,19].

This case also reflects the broader needs of *DMD* patients during transition, including coordinated multidisciplinary care, caregiver involvement, and psychological support to manage the shift to adult healthcare systems. In Romania, although limited adult neuromuscular care and fragmented services pose systemic challenges, expanding multidisciplinary collaboration and advocacy efforts create opportunities to enhance continuity and long-term outcomes.

## 2. Case Presentation

### 2.1. Methodology

We present the case of a 17-year-old patient with *DMD* who underwent transition to adult care. Retrospective data collection included clinical, paraclinical, and multidisciplinary assessments. The patient was monitored according to international care standards, with regular evaluations in neurology, cardiology, pulmonology, endocrinology, orthopedics, rehabilitation, and psychiatry.

Transition readiness was assessed at age 17 using the Transition Readiness Assessment Questionnaire (TRAQ).

### 2.2. Clinical Presentation and Diagnosis

The patient presented with mild motor impairment, noted by frequent falls and running and jumping difficulties, along with calf pseudohypertrophy, at the age of 5. Laboratory investigations revealed elevated serum levels of transaminases, lactate dehydrogenase, and creatine kinase (CK) (13,198 UI/L, with a normal reference range of 0–190 UI/L). The muscle biopsy at age 5 confirmed the complete absence of dystrophin (Dystrophin 1, 2, 3) with utrophin overexpression, as shown by immunohistochemical staining.

Given the patient’s clinical presentation and significantly elevated CK levels suggestive of muscular dystrophy, we performed multiplex ligation-dependent probe amplification (MLPA) to rule out large copy number changes. Subsequently, single-gene filtered next-generation sequencing (NGS) of the *DMD* gene was conducted, identifying a nonsense pathogenic variant, NM_004006.3(*DMD*):c.4213C>T (p.Gln1405Ter), in exon 30. This variant was classified as pathogenic according to American College of Medical Genetics and Genomics (ACMG) criteria (PVS1—Pathogenic Very Strong 1; PM2—Pathogenic Moderate 2; PS4—Pathogenic Strong 4). The results are illustrated in Figure 1.

### 2.3. Multidisciplinary Management

Given the complexity of *DMD* and its comorbidities, the patient received multidisciplinary care following international guidelines.

Initial management included continuous prednisone therapy at the age of 5, which was later adjusted to an alternate-day regimen (0.6 mg/kg/day) after two years due to side effects (weight gain and cushingoid features). At age 12, ataluren was initiated (40 mg/kg/day) as a read-through therapy, allowing the ribosome to bypass the premature stop codon and produce a partially functional dystrophin protein, thereby helping preserve skeletal muscle function.

Motor function was assessed biannually using standardized *DMD* scales. By age 15, the patient became non-ambulant, presenting with equinovarus deformity, scoliosis, and joint contractures, necessitating powered wheelchair use and assistance for positional transitions. Rehabilitation strategies were adapted, including stretching, postural support, and respiratory therapy to strengthen the cough reflex using assistive devices.

At the age of 10, an angiotensin-converting enzyme (ACE) inhibitor was initiated as a cardiac protective measure (4). During the same hospitalization, 24 h ambulatory blood pressure monitoring (ABPM) diagnosed hypertension, leading to an increase in the ACE inhibitor dose from 2 mg to 4 mg. Beta-blockers were subsequently added at age 16 to manage tachycardia. Serial cardiological evaluations revealed mild left ventricular dysfunction and a right bundle branch block.

At age 14, respiratory dysfunction was documented, with spirometry showing a forced vital capacity (FVC) of 61% predicted. Polysomnography confirmed obstructive sleep apnea, leading to the initiation of nocturnal BiLevel Positive Airway Pressure (BiPAP) ventilation, which was chosen over continuous positive airway pressure due to the presence of nocturnal hypoventilation and reduced inspiratory muscle strength, requiring pressure support for both inhalation and exhalation [20].

Renal evaluations identified elevated urea, creatinine, and cystatin C levels, requiring regular monitoring. The exact cause remains unclear but is likely multifactorial, involving chronic medication use, inflammation, and muscle breakdown.

Prolonged corticosteroid therapy resulted in obesity, iatrogenic Cushing’s syndrome, and osteoporosis. Osteoporosis was identified at age 15 and monitored over time using Dual X-ray Absorptiometry (DXA) and spinal X-rays performed with an EOS imaging system. Management included dietary interventions, calcium and vitamin D supplementation, with zoledronic acid therapy initiated at age 17.

Psychological assessments indicated an IQ of 120 but revealed social isolation, anxiety, and reluctance toward psychotherapy. Psychiatric evaluations addressed comorbidities such as anxiety and adaptation challenges related to increasing functional limitations. In response to these concerns, a multidisciplinary discussion was held involving the neurologist, psychiatrist, and psychologist, together with the patient and his family. Despite reiterating the potential benefits of psychological support, the patient declined further intervention.

### 2.4. Transition Process

Efforts were made to ensure a seamless transition by facilitating joint consultations between the pediatric and adult neurology teams, which underscored the necessity of a well-structured transition plan addressing the multifaceted needs of patients with advanced *DMD*, including specialized care for cardiac and pulmonary health, nutritional support, rehabilitation, endocrinological, and psychological well-being. The transition plan also included targeted education on emergency care.

The patient was referred to support groups for individuals with neuromuscular disorders to facilitate adaptation to adult care, providing a structured environment for peer support, social inclusion, and shared experiences in managing the challenges associated with the transition process.

At age 17, transition readiness was assessed using the Transition Readiness Assessment Questionnaire (TRAQ), which evaluates self-management skills across five key domains: medication management, appointment adherence, health monitoring, communication with healthcare providers, and self-advocacy [8,21].

The patient demonstrated strengths in healthcare communication but moderate confidence in self-management tasks; the complete results are presented in Table 1.

### 2.5. Patient Perspective

The following excerpt presents the patient’s perspective in his own words, providing direct insight into the challenges faced during transition:

“I find managing my medication on my own quite challenging. My mother handles everything related to prescriptions, purchasing, and managing potential side effects. It’s simply not something I can manage independently. Scheduling and attending appointments have become more difficult since transitioning from pediatric to adult care. While I don’t face issues with transportation, as we have a personal car, organizing consultations and ensuring all necessary tests are completed can still be complicated.When speaking with doctors, I feel confident in their expertise and explanations. If I don’t fully understand something, I always ask for additional clarification, making sure it’s explained in a way that I can grasp. I feel very involved in decisions regarding my health, but I’m not confident I could manage everything on my own. This is mainly due to the mobility challenges I face, which make many aspects of daily life, including healthcare, hard to navigate independently.The biggest difficulty I’ve encountered is finding the right specialists for my condition. To make the transition smoother, I think there should be dedicated teams of specialists who are well-informed about Duchenne Muscular Dystrophy and able to provide comprehensive care for patients like me.These insights reflect my experiences and highlight the importance of creating a better, more supportive transition process for patients with *DMD*.”

This firsthand account underscores the need for structured, multidisciplinary transition programs that integrate medical coordination with patient autonomy, ensuring continued access to specialized care and support.

## 3. Discussion

The process of transition involves a systematic progression from pediatric-focused to adult-centered healthcare systems that should be designed for adolescents and young adults suffering from chronic illnesses [10,22,23]. Thus, transition is a complex process spanning several years, requiring a multidisciplinary approach rather than a transfer between specialists.

Inadequate planning or delays in the transition process can lead to gaps in care and an incomplete handover to adult healthcare services [2,5,8,10,24,25,26,27,28]. Recent advances have culminated in an international Delphi consensus on the transition of patients with Duchenne Muscular Dystrophy from pediatric to adult care [7], which provides a unified framework for transition planning. Despite this progress, challenges remain in implementing dedicated adult care teams and ensuring seamless continuity of care.

### 3.1. Multidisciplinary Team

Although neurological management is pivotal during pediatric care for *DMD*, following transition, the emphasis shifts—cardiac and respiratory management become central to long-term survival, while the neurological focus assumes a relatively secondary role. In our patient, transition planning emphasized proactive cardiac management through regular speckle tracking echocardiographic assessments and coordinated care between pediatric and adult cardiology teams [2,5,8,11,29,30,31,32].

Specific stop codon variants have been linked to earlier motor decline and severe musculoskeletal complications such as scoliosis, contractures, and accelerated weakness [1,15,19,33,34]. Our patient lost ambulation by age 15, developing scoliosis and equinovarus deformity, and required ongoing physiotherapy, orthopedic care, and assistive technologies to maintain functional independence. In addition, chronic corticosteroid use resulted in iatrogenic Cushing’s syndrome and osteoporosis, necessitating bisphosphonate therapy [1,4,35]. We also provided the patient and his family with education on adrenal crisis management.

Our patient required dietary adjustments and gastroprotective therapy. Transition plans must ensure access to nutritional and gastroenterological support tailored to individual needs [1,4,35].

CNS dystrophin isoforms, including Dp427, Dp71, and Dp140, are associated with intellectual disabilities, anxiety, ADHD, autism spectrum disorder, and behavioral challenges, contributing to severe phenotypes [9,17,36,37,38]. It is noteworthy that our patient’s specific nonsense mutation, NM_004006.3(*DMD*):c.4213C>T (p.Gln1405Ter), introduces a premature stop codon in exon 30, which truncates the full-length dystrophin protein (Dp427). The loss of Dp427 in the brain is linked to impaired neuronal connectivity, deficits in cognitive processing, and anxiety disorders [39]. However, because the shorter isoforms (Dp140 and Dp71) are transcribed from alternative promoters located downstream of exon 30, they are generally not affected by variants in this region. These observations suggest that the psychiatric symptoms observed in our patient may be attributable not only to the progressive nature of *DMD* and its social context, but also to this specific genetic alteration. Transition care should incorporate regular psychological assessments, emotional support, and family-centered interventions to address the psychosocial impact on both patients and caregivers [2,8,9,40,41,42,43].

### 3.2. Challenges and Opportunities in Transition Care for Romanian DMD Patients

Transitioning care for patients with *DMD* in Romania presents unique challenges alongside opportunities for improvement. Systemic barriers, including fragmented healthcare systems and a shortage of multidisciplinary teams for adult patients, remain significant obstacles. However, initiatives to strengthen care pathways and promote collaboration between pediatric and adult services offer a promising path toward better outcomes.

The transition process should ideally begin between ages 12–14 and be completed by adulthood, addressing both medical and psychological needs [7,8,9]. This gradual process must maintain established care practices without imposing additional burdens on patients and their families [2,8]. Multidisciplinary evaluations, conducted in clinics with access to multiple medical specialties, are essential to reduce the need for multiple trips and streamline care coordination. These evaluations should include specialists in cardiology, pulmonology, orthopedics, endocrinology, and mental health to ensure cohesive, individualized care plans [8,11,27].

Therapies such as ataluren, which target nonsense pathogenic variants in the *DMD* gene, are available for eligible children and adults through national healthcare programs. However, ensuring continuity of therapy during the transition to adult care requires careful planning to prevent gaps in treatment adherence and monitoring [2,5,8,9]. In Romania, ataluren remains approved; however, its reimbursement for treatment-naïve patients is still under review, as the Romanian Health Insurance Fund, together with the National Association for Medicines and Medical Devices, has not yet issued a definitive statement. The European Medicines Agency did not renew the marketing authorization of ataluren, citing a lack of evidence on the risk-benefit balance [44].

Despite the lack of an integrated electronic health record (EHR) system in Romania, which complicates the transfer of essential medical information, existing practices such as thorough multidisciplinary evaluation visits and the preparation of comprehensive medical letters that clearly convey the patient’s complete history help mitigate logistical challenges. Establishing a centralized EHR system could further improve communication between healthcare providers, streamline patient management, and prevent fragmented care [8,9,11,27].

In Romania, patients with *DMD* are provided with a disability certificate and are enrolled in the National Treatment Program for Rare Diseases, which ensures reimbursement for specific medications and support services. National guidelines, established by the Ministry of Health and the National Health Insurance House, regulate insurance coverage modifications and support medical decision-making. For patients facing cognitive challenges, formal guardianship can be obtained through legal procedures. Although these frameworks are still evolving, they provide an essential foundation for coordinated, continuous care for *DMD* patients, with active involvement from patient associations to enhance access to comprehensive support.

The availability of specialists experienced in managing adult *DMD* cases is another area for development. Training programs for adult neurologists and subspecialists, alongside improved collaboration with pediatric teams, could bridge this gap and ensure continuity of care [5,8,9].

Encouraging family involvement in decision-making promotes autonomy and aligns with international recommendations for structured transitions, supporting both patients and caregivers [8,9,11].

Patient associations in Romania have become increasingly active in supporting the transition process. These organizations provide vital assistance by navigating the healthcare system, securing assistive devices, and offering emotional and informational support. Their involvement ensures that patients’ voices are heard and fosters a patient-centered approach to care [8,9,11].

Transition plans should also address broader socio-economic factors, including financial support, housing adaptations, transportation solutions, and access to assistive technologies, all of which are critical to fostering patient independence. Encouraging vocational planning, social integration, and educational opportunities tailored to the patient’s abilities can enhance quality of life and societal participation. For instance, in our case, the patient was supported in obtaining a powered wheelchair and encouraged to pursue further education through tailored plans balancing academic aspirations with medical needs, fostering intellectual growth and social inclusion [1,8,9,11,27].

While challenges persist in Romania’s transition care for *DMD*, targeted reforms, collaboration among healthcare providers, and the active involvement of patient associations create a pathway to improved outcomes. By prioritizing continuity of care, equitable access to resources, and comprehensive psychosocial support, the healthcare system can better address the unique needs of individuals with *DMD* and their families [9].

## 4. Conclusions

We presented the case of a 17-year-old Romanian patient with Duchenne Muscular Dystrophy with a nonsense variant in exon 30, highlighting the multifaceted challenges in managing this complex disorder and transitioning to adult care.

Effective management requires a multidisciplinary approach addressing neuromuscular, cardiac, respiratory, orthopedic, nutritional, endocrinological, and psychosocial needs, supported by advances in therapies and emerging treatments.

Recent international consensus guidelines emphasize that transition planning should begin by age 14 to ensure continuity of care, realistic goal setting, and proper preparation for adulthood. Our case reinforces that a structured, patient-centered protocol is essential to optimize long-term outcomes, with the patient and their family remaining at the heart of the process.

## Figures and Tables

**Figure 1 healthcare-13-00830-f001:**
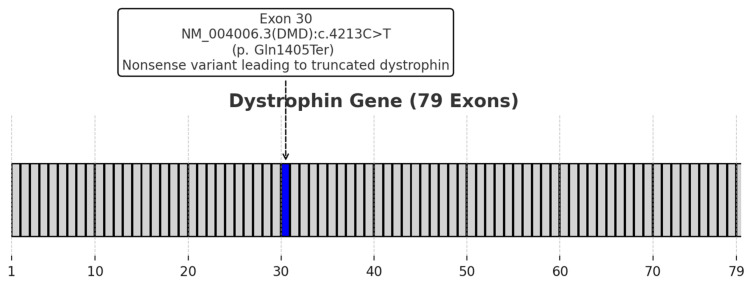
Schematic representation of the dystrophin gene (79 exons), highlighting the nonsense variant identified in exon 30 (NM_004006.3(*DMD*):c.4213C>T, p.Gln1405Ter).

**Table 1 healthcare-13-00830-t001:** Transition readiness assessment questionnaire (TRAQ) results.

TRAQ Domain	Assessment	Findings
Medication Management	Moderate	Needed caregiver assistance for refilling prescriptions and recognizing adverse effects.
Appointment Adherence	Moderate	Required reminders for scheduling and attending medical visits.
Health Monitoring	Low to moderate	Needed guidance on recognizing symptoms that warrant medical attention.
Healthcare Communication	High	Effectively discussed medical needs with providers.
Self-Advocacy and Decision-Making	Moderate	Expressed some concerns about transitioning to independent healthcare.

## Data Availability

The original data presented in this study are included in this article.

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
