# Peer review of "Lost in Transition: Challenges in the Journey from Pediatric to Adult Care for a Romanian DMD Patient"

_healthcare, 2025, doi:10.3390/healthcare13070830_

Round 1
Reviewer 1 Report
Comments and Suggestions for Authors
Dear Authors,
Thank you for the opportunity to review your manuscript. This is an important topic and many of the issues pointed out in Romania are the same globally. The adult system is not prepared for multispeciality care and that appears to be a universal issue. I think that this manuscript is important but as written it will be very challenging to follow for anyone who is not familar with DMD and this journal has a wider audience.
Genetics- you allude to the genetics but do not specifically state the mutation, I am assuming it is a single exon deletion but if it is a point mutation then exon 30 is not sufficient information.
Introduction: You should talk more broadly DMD, the outcomes and the needs for multidisciplinary care. Alternatively, briefly introduce DMD and that it represents a wide spectrum of severity across multiple system then describe the specifics common to your patients particular genotype to paint the picture of the number of specialities and comorbidity tracking involved.
Case Presentation: This contains a lot of information that is out of order and there is a lot of ancillary information that is not relevant to transition (i.e. mobility assessments performed prior to loss of ambulation when patient is currently non-ambulatory). Many therapies are introduced without or with very minimal explanation. Example Ataluren and BiPAP (i.e. why not CPAP for readers who deal with traditional OSA). Either break out the presentation chronologically or by body system - right now its mixed.
Discussion - there is a lot of ancillary information that isnt relevant (6MWT in a nonambulatory patient?) You have a mix of what might happen and what is comorbid in the patient - this needs to be only specific to patient. Does the patients mutation demonstrably affect the dystrophin isoforms? You mention the need for pysch intervention after stating that the paitnet does not want it - explain in more detail what would need to happen. What are the rules in Romania for insurance changes, guardianship, support decision-making and medical directives especially given the patient's hesitancy and case management needs.
Comments on the Quality of English LanguageThere are some typos that alter meaning for example:
Lin 41 – change genes to gene
Line 76 – cite guidelines
Line 77 : change to: angiotensin-converting enzyme (ACE) inhibitors
But overall it reads fine - its the order of the information that makes it a challenge and it reads like a bullet point list rather than a case history and story.
Author Response
Dear Reviewer 1,
Thank you very much for your useful insights on our paper! We appreciate the time and effort you have dedicated to providing feedback on our manuscript. We have incorporated changes to reflect your suggestions and have highlighted them in yellow within the manuscript.
Comment 1: Genetics- you allude to the genetics but do not specifically state the mutation, I am assuming it is a single exon deletion but if it is a point mutation then exon 30 is not sufficient information.
Response 1: The exact variant has now been specified, and we have clarified its classification as pathogenic based on the ACMG criteria (PVS1, PM2, PS4).
Additionally, Figure 1 now illustrates this genetic finding for clarity.
The revised text now reads:
“Given the patient’s clinical presentation and significantly elevated CK levels suggestive of muscular dystrophy, we performed multiplex ligation-dependent probe amplification (MLPA) to rule out large copy number changes. Subsequently, single-gene filtered next-generation sequencing (NGS) of the DMD gene was conducted, identifying a nonsense pathogenic variant, NM_004006.3(DMD):c.4213C>T (p.(Gln1405Ter)), in exon 30. This variant was classified as pathogenic according to American College of Medical Genetics and Genomics (ACMG) criteria (PVS1—Pathogenic Very Strong 1, PM2—Pathogenic Moderate 2, PS4—Pathogenic Strong 4). The results are illustrated in Figure 1.”
Figure 1. Schematic representation of the dystrophin gene (79 exons), highlighting the nonsense variant identified in exon 30 (NM_004006.3(DMD):c.4213C>T, p.(Gln1405Ter))
Comment 2: Introduction: You should talk more broadly DMD, the outcomes and the needs for multidisciplinary care. Alternatively, briefly introduce DMD and that it represents a wide spectrum of severity across multiple system then describe the specifics common to your patients particular genotype to paint the picture of the number of specialties and comorbidity tracking involved.
Response 2: Thank you for your valuable feedback. We have revised the introduction to emphasize DMD's multisystem involvement, the necessity for a comprehensive multidisciplinary approach, and the importance of robust, individualized transition planning—especially given the challenges encountered in Romania.
The revised text is provided below:
“Dystrophinopathies are a group of neuromuscular disorders caused by variants in dystrophin-encoding gene, leading to absent or deficient dystrophin production and progressive muscle degeneration [1,2]. Duchenne Muscular Dystrophy (DMD) (OMIM #310200; ORPHA:98896), an X-linked genetic disorder, is the most common and severe form of muscular dystrophy [1,3]. Beyond its primary neuromuscular involvement, DMD is a multisystemic disorder that affects the cardiovascular, respiratory, gastrointestinal, endocrine, and central nervous systems [1,4,5). Given the extensive nature of its clinical manifestations, effective management necessitates a multidisciplinary approach [1,4–6].
Since the 1990s, the introduction of steroid therapy, the broader use of ventilation support, comprehensive multidisciplinary care, and the protective administration of angiotensin-converting enzyme (ACE) inhibitors have all contributed to a marked increase in life expectancy, with many patients now living well beyond 40 years of age [1,2,6–8].
With ongoing improvements in care and extended life expectancy, the transition from pediatric to adult-focused medicine has become essential for all DMD patients. A structured transition plan that anticipates individual complexity and needs remains crucial to maintaining high standards of care [2,7,9–11].
Nonsense variants, which comprise around 10–15% of DMD cases, introduce a premature stop codon leading to a truncated dystrophin protein [12–14]. Their clinical impact varies based on the stop codon’s location (15), but certain positions have been linked to earlier onset of cardiomyopathy, cognitive impairments, and accelerated muscle deterioration [16–18]. Such genotypic differences highlight the importance of tailored multidisciplinary management, ensuring each patient’s unique profile is addressed across specialties [5,15–19].
This case also reflects the broader needs of DMD patients during transition, including coordinated multidisciplinary care, caregiver involvement, and psychological support to manage the shift to adult healthcare systems. In Romania, although limited adult neuromuscular care and fragmented services pose systemic challenges, expanding multidisciplinary collaboration and advocacy efforts create opportunities to enhance continuity and long-term outcomes.”
Comment 3: Case Presentation: This contains a lot of information that is out of order and there is a lot of ancillary information that is not relevant to transition (i.e. mobility assessments performed prior to loss of ambulation when patient is currently non-ambulatory).
Response 3: Thank you for your valuable feedback. We have revised the case presentation by removing redundant and ancillary information not directly related to the transition process. We reorganised and condensed the content to improve clarity and ensure a more coherent and easily navigable narrative.
Comment 4: Many therapies are introduced without or with very minimal explanation. Example Ataluren and BiPAP (i.e. why not CPAP for readers who deal with traditional OSA).
Response 4: We have provided further details, clarifying that Ataluren was selected due to its mechanism as a readthrough therapy targeting nonsense mutations, while BiPAP was chosen over CPAP due to the presence of nocturnal hypoventilation and reduced inspiratory muscle strength. The text now reads:
„At age 12, Ataluren was initiated to target the nonsense pathogenic variant (40 mg/kg/day).”
„Polysomnography confirmed obstructive sleep apnea, leading to the initiation of nocturnal BiLevel Positive Airway Pressure (BiPAP) ventilation which was chosen over continuous positive airway pressure due to the presence of nocturnal hypoventilation and reduced inspiratory muscle strength, requiring pressure support for both inhalation and exhalation (16).”
Comment 5: Either break out the presentation chronologically or by body system - right now its mixed.
Response 5: We have revised the case presentation to improve clarity and organization. To enhance readability and logical flow, we have restructured it as follows: 2.1. Methodology, 2.2. Clinical presentation and diagnosis, 2.3. Multidisciplinary management, 2.4. Transition process and 2.5. Patient perspective. Furthermore, we have added a table summarizing the TRAQ (Transition Readiness Assessment Questionnaire) results to provide a clearer overview of the patient’s transition readiness assessment.
Comment 6: Discussion - there is a lot of ancillary information that isn’t relevant (6MWT in a nonambulatory patient?)
Response 6: Thank you for your comment. We have removed the redundant information.
Comment 7: You have a mix of what might happen and what is comorbid in the patient - this needs to be only specific to patient.
Response 7: We have revised the discussion to include only information specific to our patient, removing generalized statements and focusing solely on his observed comorbidities.
Comment 8: Does the patient’s mutation demonstrably affect the dystrophin isoforms?
Response 8: This particular variant, NM_004006.3(DMD):c.4213C>T (p.(Gln1405Ter)), introduces a premature stop codon in exon 30 that results in the truncation of the full-length dystrophin protein (Dp427). However, because the shorter dystrophin isoforms—such as Dp140 and Dp71—are transcribed from alternative promoters located downstream of exon 30, they are generally not affected by variants in this region. These observations suggest that the psychiatric symptoms observed in our patient may be attributable not only to the progressive nature of DMD and its social context, but also to this specific genetic alteration. This revised information has been incorporated into the manuscript.
Comment 9: You mention the need for pysch intervention after stating that the patient does not want it - explain in more detail what would need to happen.
Response 9: We have expanded the discussion regarding the need for psychological intervention. The text has been revised accordingly and now reads:
“Psychological assessments indicated an IQ of 120 but revealed social isolation, anxiety, and reluctance toward psychotherapy. Psychiatric evaluations addressed comorbidities such as anxiety and adaptation challenges related to increasing functional limitations. In response to these concerns, a multidisciplinary discussion was held involving the neurologist, psychiatrist, and psychologist, together with the patient and his family. Despite reiterating the potential benefits of psychological support, the patient declined further intervention.”
Comment 10: What are the rules in Romania for insurance changes, guardianship, support decision-making and medical directives especially given the patient's hesitancy and case management needs.
Response 10: In Romania, patients with DMD are issued a disability certificate and enrolled in the National Treatment Program for Rare Diseases, which guarantees specific medical coverage and support. Insurance coverage modifications and support for decision-making are managed under national guidelines set by the Ministry of Health and the National Health Insurance House, and formal guardianship is established through legal procedures if needed. We have modified the discussion to include the following:
"In Romania, patients with DMD are provided with a disability certificate and are enrolled in the National Treatment Program for Rare Diseases, which ensures reimbursement for specific medications and support services. National guidelines, established by the Ministry of Health and the National Health Insurance House, regulate insurance coverage modifications and support medical decision-making. For patients facing cognitive challenges, formal guardianship can be obtained through legal procedures. Although these frameworks are still evolving, they provide an essential foundation for coordinated, continuous care for DMD patients, with active involvement from patient associations to enhance access to comprehensive support."
Comment 11: Line 41 – change genes to gene.
Response 11: We have corrected "genes" to "gene".
Comment 12: Line 76 – cite guidelines
Response 12: We have now cited the relevant guidelines in the text.
Comment 13: Line 77 : change to: angiotensin-converting enzyme (ACE) inhibitors
Response 13: We have made this change at the first mention in the text, replacing it with "angiotensin-converting enzyme (ACE) inhibitors" as recommended.

Reviewer 2 Report
Comments and Suggestions for Authors
The authors analyze a single case of a patient with Duchenne muscular dystrophy undergoing transition from a pediatric to an adult care center, with an interesting perspective from the patient included in the manuscript.
That said, unfortunately, the manuscript lacks many fundamental details, the most critical being the scarce numerosity, being a case report, and the update of the scientific literature (a recent consensus article using the Delphi method specifically addressing transition in Duchenne muscular dystrophy has been published, as mentioned below).
-
Lines 42-45: I recommend that the authors review the most recent literature regarding life expectancy in DMD according to the latest SOC and management of complications. Mortality has globally improved, with occasional cases exceeding 40-45 years, but overall life expectancy now often surpasses 30 years.
-
Lines 46-50: it is incorrect to state that stop-codon mutations categorically result in more severe phenotypes. This statement should be revised (e.g., Torella A, Zanobio M, Zeuli R, Del Vecchio Blanco F, Savarese M, Giugliano T, Garofalo A, Piluso G, Politano L, Nigro V. The position of nonsense mutations can predict the phenotype severity: A survey on the DMD gene. PLoS One. 2020 Aug 19;15(8):e0237803. doi: 10.1371/journal.pone.0237803. PMID: 32813700; PMCID: PMC7437896).
-
Line 48: typo in “cardiomyopathy.”
-
Lines 54-55: transition is important regardless of the type of mutation. I recommend the authors reorganize this sentence.
-
Lines 61-62: the readability of this sentence is unclear.
-
Line 64: the variant is not specified, and it is unclear how its pathogenicity was determined.
-
Lines 65-66: what alternate-day prednisone regimen was prescribed?
-
Line 67: please describe motor milestone acquisition and define what the authors meant by “mild motor impairment.”
-
Lines 68-69: LDH and transaminase levels are not as clinically relevant in DMD as CK levels. What were the CK values? Please include reference ranges for the laboratory.
-
Line 69: at what age was the muscle biopsy performed? Was it an open biopsy or a needle biopsy? What was the biopsy site? How was dystrophin absence verified? Only IHC or also Western Blot (WB)?
-
Lines 71-72: MLPA primarily detects copy number variants; was additional analysis performed using NGS panels or direct sequencing of the DMD gene?
-
Line 72: at what age did the patient start prednisone?
-
Line 73: what adverse events did the patient experience?
-
Lines 76-77: steroid-induced hypertension is not particularly common in early life in DMD. Was the ACE inhibitor initiated as cardiac protection according to international standards of care at age 10?
-
Line 90: increased creatinine levels are rare in patients with severe muscle atrophy (or were only urea and cystatin C elevated?). Did the patient misuse NSAIDs? It should be specified that this is iatrogenic damage(from which medications?), rather than due to muscle breakdown or inflammation (of what type?).
-
Line 95: DXA and spinal X-rays are monitoring tools, not interventions. They should be placed in parentheses next to “osteoporosis” as “identified with…”
-
Line 100: please specify that TRAQ was assessed at 17 years old.
-
Lines 104-105: add "endocrinological".
-
Line 118, Lines 123-124: this statement is incorrect. In January 2025, a Delphi consensus specific to Duchenne muscular dystrophy transition was published:
Castro D, Sejersen T, Bello L, Buccella F, Cairns A, Carranza-Del Río J, de Groot IJM, Elman L, Inzani I, Klein A, Mayer OH, Miller H, Onofri A, Araújo APQC, Schara-Schmidt U, Vanden Wyngaert K, Ward LM, Wilmshurst JM, Quinlivan R. Transition of patients with Duchenne muscular dystrophy from paediatric to adult care: An international Delphi consensus study. Eur J Paediatr Neurol. 2025 Jan 11;54:130-139. doi: 10.1016/j.ejpn.2025.01.004. Epub ahead of print. PMID: 39892019. -
Lines 179-185: is ataluren still available (for naïve patients) despite the EMA's unfavorable opinion?
Author Response
Dear Reviewer,
Thank you for your useful feedback on our submission.
We have carefully addressed each of your comments and suggestions. The following responses outline the revisions made to the manuscript.
Comment 1:” Lines 42-45: I recommend that the authors review the most recent literature regarding life expectancy in DMD according to the latest SOC and management of complications. Mortality has globally improved, with occasional cases exceeding 40-45 years, but overall life expectancy now often surpasses 30 years.”
Response 1: Thank you for highlighting this issue. We agree that it is an important approach. To provide the necessary context and details, we have added an introductory paragraph, which now reads: “Since the 1990s, the introduction of steroid therapy, the broader use of ventilation support, comprehensive multidisciplinary care, and the protective administration of angiotensin-converting enzyme (ACE) inhibitors have all contributed to a marked increase in life expectancy, with many patients now living well beyond 40 years of age (1,2,4–6).”
Comment 2: “Lines 46-50: it is incorrect to state that stop-codon mutations categorically result in more severe phenotypes. This statement should be revised (e.g., Torella A, Zanobio M, Zeuli R, Del Vecchio Blanco F, Savarese M, Giugliano T, Garofalo A, Piluso G, Politano L, Nigro V. The position of nonsense mutations can predict the phenotype severity: A survey on the DMD gene. PLoS One. 2020 Aug 19;15(8):e0237803. doi: 10.1371/journal.pone.0237803. PMID: 32813700; PMCID: PMC7437896)”
Response 2: We appreciate your feedback and have clarified the statement accordingly. The updated text now reads: “Nonsense variants, which comprise around 10–15% of DMD cases, introduce a premature stop codon leading to a truncated dystrophin protein [9-11]. Their clinical impact varies based on the stop codon’s location [12], but certain positions have been linked to earlier onset of cardiomyopathy, cognitive impairments, and accelerated muscle deterioration [6-8].”
Comment 3: “Line 48: typo in “cardiomyopathy.”
Response 3: We have corrected 'cardiomiopathy' to 'cardiomyopathy' in the manuscript.
Comment 4: “Lines 54-55: transition is important regardless of the type of mutation. I recommend the authors reorganize this sentence.”
Response 4: We have rephrased the sentence. The updated text now reads: “With ongoing improvements in care and extended life expectancy, the transition from pediatric to adult-focused medicine has become essential for all DMD patients. While specific variants (e.g., nonsense mutations) can present more severe phenotypes [6-8]”
Comment 5: “Lines 61-62: the readability of this sentence is unclear.”
Response 5: We have revised the sentence for clarity: “In Romania, although limited adult neuromuscular care and fragmented services pose systemic challenges, expanding multidisciplinary collaboration and advocacy efforts create opportunities to enhance continuity and long-term outcomes.”
Comment 6: ”Line 64: the variant is not specified, and it is unclear how its pathogenicity was determined.”
Response 6: Thank you for your observation. The exact variant has now been specified, and we have clarified its classification as pathogenic based on the ACMG criteria (PVS1, PM2, PS4). The revised text now reads: “Given the patient’s clinical presentation and significantly elevated CK levels suggestive of muscular dystrophy, we performed multiplex ligation-dependent probe amplification (MLPA) to rule out large copy number changes. Subsequently, single-gene filtered next-generation sequencing (NGS) of the DMD gene was conducted, identifying a nonsense pathogenic variant, NM_004006.3(DMD):c.4213C>T (p.(Gln1405Ter)), in exon 30. This variant was classified as pathogenic according to American College of Medical Genetics and Genomics (ACMG) criteria (PVS1—Pathogenic Very Strong 1, PM2—Pathogenic Moderate 2, PS4—Pathogenic Strong 4). The results are illustrated in Figure 1.”
Comment 7: “Lines 65-66: what alternate-day prednisone regimen was prescribed?”
Response 7: We’ve clarified this point — the patient was prescribed an alternate-day prednisone regimen of 0.6 mg/kg/day. The updated text now reads: “Initial management included continuous prednisone therapy at the age of 5, which was later adjusted to an alternate-day regimen (0.6 mg/kg/day) after two years due to side effects. ”Initial management included continuous prednisone therapy at the age of 5, which was later adjusted to an alternate-day regimen (0.6 mg/kg/day) after two years due to side effects (weight gain and cushingoid features).”
Comment 8: “Line 67: please describe motor milestone acquisition and define what the authors meant by “mild motor impairment.”
Response 8: We have now specified that ‘mild motor impairment’ refers to frequent falls and difficulty running and jumping, indicating subtle but notable deficits in gross motor function. The revised text now reads: „The patient presented with mild motor impairment, noted by frequent falls and running and jumping difficulties, along with calf pseudohypertrophy, at the age of 5.”
Comment 9: " Lines 68-69: LDH and transaminase levels are not as clinically relevant in DMD as CK levels. What were the CK values? Please include reference ranges for the laboratory.”
Response 9: Thank you for your suggestion. We have now included the CK value (13,198 UI/L, with a normal reference range of 0–190 UI/L) to provide clearer clinical context.
Comment 10: “Line 69: at what age was the muscle biopsy performed? Was it an open biopsy or a needle biopsy? What was the biopsy site? How was dystrophin absence verified? Only IHC or also Western Blot (WB)?”
Response 10: We appreciate your insightful questions. The muscle biopsy was performed at age 5 via an open procedure on the right gastrocnemius. Immunohistochemical staining (IHC) confirmed the complete absence of dystrophin (Dystrophin 1, 2, 3) with utrophin overexpression, while Western blot analysis was not performed. The text has been updated to: “The muscle biopsy at age 5 confirmed the complete absence of dystrophin (Dystrophin 1, 2, 3) with utrophin overexpression, as shown by immunohistochemical staining.”
Comment 11: “Lines 71-72: MLPA primarily detects copy number variants; was additional analysis performed using NGS panels or direct sequencing of the DMD gene?”
Response 11: We appreciate your question. Initially, MLPA was used to screen for large copy number variants. To further investigate, we performed single-gene filtered (NGS) of the DMD gene. This analysis identified the c.4213C>T nonsense variant in exon 30, confirming the diagnosis. We have modified the text to:
“Given the patient’s clinical presentation and significantly elevated CK levels suggestive of muscular dystrophy, we performed multiplex ligation-dependent probe amplification (MLPA) to rule out large copy number changes. Subsequently, single-gene filtered next-generation sequencing (NGS) of the DMD gene was conducted, identifying a nonsense pathogenic variant, NM_004006.3(DMD):c.4213C>T (p.(Gln1405Ter)), in exon 30. This variant was classified as pathogenic according to American College of Medical Genetics and Genomics (ACMG) criteria (PVS1—Pathogenic Very Strong 1, PM2—Pathogenic Moderate 2, PS4—Pathogenic Strong 4). The results are illustrated in Figure 1.”
Additionally, Figure 1 now illustrates this genetic finding for clarity.
Figure 1. Schematic representation of the dystrophin gene (79 exons), highlighting the nonsense variant identified in exon 30 (NM_004006.3(DMD):c.4213C>T, p.(Gln1405Ter))
Comment 12: “Line 72: at what age did the patient start prednisone?”
Response 12: We have included in the text that prednisone therapy was initiated at age 5. The updated text is now: “Initial management included continuous prednisone therapy at the age of 5, which was later adjusted to an alternate-day regimen (0.6 mg/kg/day) after two years due to side effects.”
Comment 13: “Line 73: what adverse events did the patient experience?”
Response 13: We have clarified in the text that prednisone therapy was initiated at age 5 and included details on the rationale for switching to an alternate-day regimen. The revised version states: “Initial management included continuous prednisone therapy at the age of 5, which was later adjusted to an alternate-day regimen (0.6 mg/kg/day) after two years due to side effects (weight gain and cushingoid features).”
Comment 14: “Lines 76-77: steroid-induced hypertension is not particularly common in early life in DMD. Was the ACE inhibitor initiated as cardiac protection according to international standards of care at age 10?”
Response 14: Thank you for your comment. The ACE inhibitor was initially prescribed as a cardiac protective measure, consistent with international guidelines. However, during the same hospitalization at age 10, 24-hour ambulatory blood pressure monitoring (ABPM) diagnosed hypertension, prompting an increase in the ACE inhibitor dose from 2 mg to 4 mg. The revised text now reads: "By age 10, an (angiotensin-converting enzyme) ACE inhibitor was initiated as a cardiac protective measure. During the same hospitalization, 24-hour ambulatory blood pressure monitoring (ABPM) diagnosed hypertension, leading to an increase in the ACE inhibitor dose from 2 mg to 4 mg, and beta-blockers were subsequently added at age 16 to manage tachycardia."
Comment 15: “Line 90: increased creatinine levels are rare in patients with severe muscle atrophy (or were only urea and cystatin C elevated?). Did the patient misuse NSAIDs? It should be specified that this is iatrogenic damage(from which medications?), rather than due to muscle breakdown or inflammation (of what type?)”
Response 15: We acknowledge that increased creatinine levels are uncommon in patients with severe muscle atrophy. While the exact cause remains unclear in this case, the elevation was likely multifactorial, potentially related to chronic medication use rather than solely muscle breakdown or inflammation. We have adjusted the text accordingly: "Renal evaluations identified elevated urea, creatinine, and cystatin C levels, requiring regular monitoring. The exact cause remains unclear but is likely multifactorial, involving chronic medication use, inflammation, and muscle breakdown."
Comment 16: “Line 95: DXA and spinal X-rays are monitoring tools, not interventions. They should be placed in parentheses next to “osteoporosis” as “identified with…”
Response 16: We have revised the text: “Prolonged corticosteroid therapy resulted in obesity, iatrogenic Cushing's syndrome, and osteoporosis. Osteoporosis was identified and monitored over time using Dual X-ray Absorptiometry (DXA) and spinal X-rays performed with an EOS imaging system, and it was managed with dietary interventions, calcium and vitamin D supplementation, and zoledronic acid therapy initiated at age 17."
Comment 17: “Line 100: please specify that TRAQ was assessed at 17 years old.”
Response 17: We have specified that TRAQ was assessed at 17 years of age.
Comment 18: “Lines 104-105: add ‘endocrinological’.”
Response 18: We have added 'endocrinological'.
Comment 19: “Line 118, Lines 123-124: this statement is incorrect. In January 2025, a Delphi consensus specific to Duchenne muscular dystrophy transition was published:
Castro D, Sejersen T, Bello L, Buccella F, Cairns A, Carranza-Del Río J, de Groot IJM, Elman L, Inzani I, Klein A, Mayer OH, Miller H, Onofri A, Araújo APQC, Schara-Schmidt U, Vanden Wyngaert K, Ward LM, Wilmshurst JM, Quinlivan R. Transition of patients with Duchenne muscular dystrophy from paediatric to adult care: An international Delphi consensus study. Eur J Paediatr Neurol. 2025 Jan 11;54:130-139. doi: 10.1016/j.ejpn.2025.01.004. Epub ahead of print. PMID: 39892019.”
Response 19: At the time of manuscript finalization, the January 2025 Delphi consensus was not yet available. We appreciate your input and have updated the text to reflect these recent developments. The text now reads: "Inadequate planning or delays in the transition process can lead to gaps in care and an incomplete handover to adult healthcare services (2,15–22). Recent advances have culminated in an international Delphi consensus on the transition of patients with Duchenne muscular dystrophy from pediatric to adult care [4], which provides a unified framework for transition planning. Despite this progress, challenges remain in implementing dedicated adult care teams and ensuring seamless continuity of care.”
Comment 20: “Lines 179-185: is ataluren still available (for naïve patients) despite the EMA's unfavorable opinion?”
Response 20: At the time of writing, the final decision on Ataluren's availability for treatment-naïve patients was still pending. The Romanian Health Insurance House and the National Agency for Medicines and Medical Devices had not yet issued a definitive statement, despite the EMA's unfavorable opinion.
"In Romania, the final decision regarding the availability of Ataluren is still pending, as the Romanian Health Insurance House, together with the National Agency for Medicines and Medical Devices, has not yet provided a definitive statement, despite the EMA's unfavorable opinion [44]."
Thank you for your insightful comments. Considering your suggestions, we have revised the conclusions:
”We presented the case of a 17-year-old Romanian patient with Duchenne muscular dystrophy carrying a nonsense variant in exon 30, highlighting the multifaceted challenges of managing this complex disorder and transitioning to adult care. Effective management requires a multidisciplinary approach that addresses neuromuscular, cardiac, respiratory, orthopedic, nutritional, endocrinological, and psychosocial needs, supported by advances in therapies and emerging treatments. Recent international consensus guidelines emphasize that transition planning should begin by age 14 to ensure continuity of care, realistic goal setting, and proper preparation for adulthood. Our case reinforces that a structured, patient-centered protocol is essential for optimizing long-term outcomes, with the patient and their family remaining at the heart of the process.”

Round 2
Reviewer 1 Report
Comments and Suggestions for Authors
Dear Authors,
Thank you for the opportunity to review this revised manuscript. Your edits have made the flow and manuscript intent much clearer. I have some minor recommendations mostly targeting readers who are likely unfamiliar with DMD.
Line 92 - remove parentheses inside —> p.Gln1405Ter
Readers may not be familiar with DMD
Line 102 - state for skeletal muscle preservation
Line 104 - state that Ataluren is a read-through therapy that allows the mRNA to create a functional protein
Line 111 - angiotensin-converting enzyme (ACE) inhibitors
Line 112 - what was he hospitalized for? Same meaning age 10 or age 16?
Line 130 - all osteo meds introduced at age 17? If yes leave, if no maybe indicate age that osteoporosis was identified.
Section 2.5 - is this a direct quote? Maybe introduce with a sentence and then indent as a long quotation then close with your insights sentence
Line 239 - 243 What does pending availability mean?
Line 242 - change to European Medicines Agency did not renew the marketing authorization of ataluren citing lack of evidence on risk-benefit balance.
Of note, the EMA lists ataluren as the generic name so it should be lower case in your manuscript also.
Thank you!
Author Response
Dear Reviewer,
Thank you for dedicating your time to review our manuscript. We have incorporated changes based on your suggestions, and these are highlighted in yellow within the text. Below, please find our detailed responses to each of your comments.
Comment 1: Line 92 - remove parentheses inside —> p.Gln1405Ter. Readers may not be familiar with DMD
Response 1: We have removed the parentheses.
Comment 2: Line 102 - state for skeletal muscle preservation
Line 104 - state that Ataluren is a read-through therapy that allows the mRNA to create a functional protein
Response 2: Thank you for your suggestion. We have updated the text. It now reads: “At age 12, ataluren was initiated (40 mg/kg/day) as a read-through therapy, allowing the ribosome to bypass the premature stop codon and produce a partially functional dystrophin protein, thereby helping preserve skeletal muscle function”.
Comment 3: Line 111 - angiotensin-converting enzyme (ACE) inhibitors
Response 3: We have modified in the text.
Comment 4: Line 112 - what was he hospitalized for? Same meaning age 10 or age 16?
Response 4: The hospitalization at age 10 was part of the patient’s routine periodic evaluation. During that admission, an ACE inhibitor was initiated as a cardiac protective measure, and ABPM revealed hypertension—leading to an increased dose. Beta-blockers were later introduced at age 16 to address tachycardia. We have clarified this in the text to avoid confusion. The text reads:
“At the age of 10, an angiotensin-converting enzyme (ACE) inhibitor was initiated as a cardiac protective measure (4). During the same hospitalization, 24-hour ambulatory blood pressure monitoring (ABPM) diagnosed hypertension, leading to an increase in the ACE inhibitor dose from 2 mg to 4 mg. Beta-blockers were subsequently added at age 16 to manage tachycardia.”
Comment 5: Line 130 - all osteo meds introduced at age 17? If yes leave, if no maybe indicate age that osteoporosis was identified.
Response 5: Thank you for your observation. We have clarified the timeline by specifying when osteoporosis was identified. The revised text now reads:
"Prolonged corticosteroid therapy resulted in obesity, iatrogenic Cushing's syndrome, and osteoporosis. Osteoporosis was identified at age 15 and monitored over time using Dual X-ray Absorptiometry (DXA) and spinal X-rays performed with an EOS imaging system. Management included dietary interventions, calcium and vitamin D supplementation, with zoledronic acid therapy initiated at age 17."
Comment 6: Section 2.5 - is this a direct quote? Maybe introduce with a sentence and then indent as a long quotation then close with your insights sentence
Response 6: Thank you for your suggestion. We have revised Section 2.5 to introduce the patient's statement and format it as a block quote. Additionally, we included a closing reflection to reinforce the significance of the patient’s experience in guiding transition improvements.
The updated text now reads:
"The following excerpt presents the patient’s perspective in his own words, providing direct insight into the challenges faced during transition:"
[Indented block quote containing the patient’s statement]
"This firsthand account underscores the need for structured, multidisciplinary transition programs that integrate medical coordination with patient autonomy, ensuring continued access to specialized care and support."
Comment 7: Line 239 - 243 What does pending availability mean?
Line 242 - change to European Medicines Agency did not renew the marketing authorization of ataluren citing lack of evidence on risk-benefit balance.
Response 7: We have revised the text accordingly. The updated sentence now reads:
"In Romania, Ataluren remains approved; however, its reimbursement for treatment-naïve patients is still under review, as the Romanian Health Insurance Fund together with the National Association for Medicines and Medical Devices have not yet issued a definitive statement. The European Medicines Agency did not renew the marketing authorization of Ataluren, citing a lack of evidence on the risk-benefit balance (44)."
Comment 8: Of note, the EMA lists ataluren as the generic name so it should be lower case in your manuscript also.
Response 8: Thank you for your observation. We have updated the manuscript accordingly, ensuring that "ataluren" is written in lowercase throughout.

Reviewer 2 Report
Comments and Suggestions for Authors
The authors have improved the quality and content of the article as recommended. However, it remains limited in terms of impact, as it is a case report. As previously suggested, it would be advisable to collect at least a series of cases before proceeding with publication to enhance the significance of the work.
Author Response
Dear Reviewer,
Thank you for taking the time to review our manuscript and for sharing your insights!
Comment: The authors have improved the quality and content of the article as recommended. However, it remains limited in terms of impact, as it is a case report. As previously suggested, it would be advisable to collect at least a series of cases before proceeding with publication to enhance the significance of the work.
Response: Expanding our analysis to a series of cases would effectively result in an entirely different article, requiring a larger scope and methodology. Our focused, in-depth examination of this single patient’s transition journey offers insights into systemic barriers faced by many DMD patients in Romania. We hope this targeted approach will be considered a valuable contribution to the literature on transitional care in DMD.
